# Opportunities and Challenges in Doubled Haploids and Haploid Inducer-Mediated Genome-Editing Systems in Cucurbits

**Isidre Hooghvorst [1,2,*]** and **Salvador Nogués [1]**

[1] Departament de Biologia Evolutiva, Ecologia i Ciencies Ambientals, Secció de Fisiologia Vegetal, Universitat de Barcelona, 08028 Barcelona, Spain; salvador.nogues@ub.edu

[2] Rocalba S.A, 17002 Girona, Spain

[*] Correspondence: isidrevander@gmail.com

**Abstract:** Doubled haploids have played a major role in cucurbit breeding for the past four decades. In situ parthenogenesis via irradiated pollen is the preferred technique to obtain haploid plantlets whose chromosomes are then doubled in Cucurbitaceae, such as melon, cucumber, pumpkin, squash and winter squash. In contrast to doubled haploid procedures in other species, in situ parthenogenesis in cucurbits presents many limiting factors which impede efficient production of haploids. In addition, it is very time-consuming and labor-intensive. However, the haploid inducer-mediated genome-editing system is a breakthrough technology for producing doubled haploids. Several reports have described using the CRISPR/Cas9 system in cucurbit species, and although its application has many bottlenecks, the targeted knock-out of the CENH3 gene will allow breeders to obtain haploid inducer lines that can be used to obtain parthenogenetic embryos. In this review, we discuss the progress made towards the development of doubled haploids and haploid inducer genotypes using CRISPR/Cas9 technologies in cucurbit species. The present review provides insights for the application of haploid inducer-mediated genome-editing system in cucurbit species

**Keywords:** cucurbits; doubled haploids; parthenogenesis; genome editing; CRISPR/Cas9; haploid-inducer

## 1. Introduction

The Cucurbitaceae is a family with several important crop species and contains a great genetic, morphologic and phenotypic variability. In terms of worldwide production, the most important species are watermelon with 104 m tons, cucumber and gherkins with 75 m tons, pumpkin, squash and gourds with 28 m tons and melon with 24 m tons [1]. Commercial cultivars are usually F1 hybrids due to heterosis, which results in earlier harvest, increased yield and higher vigor [2].

Therefore, pure lines used as parents for hybrid production are invaluable. There are two main methods in cucurbits to produce completely homozygous lines, classical breeding and doubled haploids. Classic breeding requires several rounds of selfing and selection for eight to ten generations, which consumes a considerable amount of time and resources. On the other hand, doubled haploid (DH) methodology allows the efficient production of completely homozygous pure lines in less than two years. Parthenogenic DH methodology in cucurbits consists of two basic steps, the initial production of haploid material through in situ parthenogenesis via pollination with irradiated pollen and the subsequent chromosome doubling of haploid plants to restore the diploid chromosome content and to allow the generation of DH seed. Nevertheless, parthenogenesis in cucurbits has

several bottlenecks and limitations that may jeopardize the final production of DH lines, such as high genotypic dependency; low fruit set when pollinated with irradiated pollen; difficulties in detecting parthenogenetic embryos; low production of parthenogenetic embryos; recalcitrant culture in vitro; high mortality of parthenogenetic plants during in vitro culture and after the chromosome doubling treatment; low rates of chromosome doubling; and low fruit set of DH lines once chromosomes are doubled [3].

DHs have been widely used to obtain stabilized homozygous lines to be used as parental lines to produce F1 hybrids. On the other hand, stabilized lines have been used for: establishing chromosome maps and whole genome; bulked segregant analysis (BSA); and, for mapping quantitative trait loci (QTLs) [4].

Until recently, classical breeding and DHs were the main pathways for the production of parental lines. However, the haploid inducer-mediated genome-editing system is a breakthrough approach that could lead to a new era of cucurbit breeding. The currently-predominant genome editing technique is CRISPR/Cas9 due to its efficiency, versatility, and unprecedented control over mutation. CRISPR/Cas9 consists of a Cas9 nuclease guided by a 20-nt sequence (gRNA). This system induces DNA double-strand breaks (DSBs) that are repaired by either non-homologous end-joining (NHEJ) or homology directed repair (HDR), generating insertion and deletion events (INDELs) [5].

There are numerous reports and reviews of DH production and CRISPR/Cas9 applications in many plant species including cucurbits. Nevertheless, haploid inducer-mediated genome-editing systems have not yet been applied in cucurbits. Therefore, the purpose of this review is to focus on haploid inducer-mediated genome-editing systems in cucurbit species to provide new insights, opportunities and challenges, which may be valuable for developing this technique in cucurbits and other species.

## 2. Doubled Haploids Procedure in Cucurbits

Doubled haploid lines in cucurbit species can be produced by parthenogenesis, androgenesis or gynogenesis. However, parthenogenesis is the currently-predominant technique to produce doubled haploids in cucumber, melon, watermelon and pumpkins [6]. The parthenogenesis process starts with in vivo pollination using irradiated pollen. The resulting parthenogenetic embryos are then detected, rescued and cultured in vitro, in order to germinate and develop into plantlets. The ploidy-level of parthenogenetic plantlets is estimated by flow cytometry and can result in haploid, spontaneous doubled haploid, mixoploid or tetraploid levels. The haploid and mixoploid plantlets need to undergo in vitro or in vivo chromosome doubling, usually using colchicine as antimitotic compound. Once the plantlets have doubled their chromosomes, they are cultured in the greenhouse together with the spontaneously-doubled haploid lines to recover DH seed [3,6,7]. During the DH process, a high genotypic dependency and other factors continuously hamper each step causing a loss of efficiency that might be critical.

### 2.1. Pollination with Irradiated Pollen and Fruit Set

The first step of the in situ parthenogenetic process is the irradiation of mature pollen and the pollination of the receptor female flowers. $\gamma$-ray ($^{60}$Co or $^{137}$Cs) and soft X-ray are the usual irradiation sources applied to male flowers. The ionized pollen can germinate on the female stigma and grow pollen tubes to reach the embryo sac. However, this pollen is genetically inactive and unable to fertilize the egg-cell and the polar nuclei. Therefore, irradiated pollen stimulates egg-cell division and parthenogenetic embryo induction [3]. Overall, the dose of ionizing radiation can range from 25 to 500 Gy, depending on the species and can yield less parthenogenetic haploid embryos at higher or more diploid embryos at lower dosages. Therefore, irradiation should be optimized for each species because pollen sensitivity is attributed to radiation-resistance.

The parameters that define the success or failure of pollination with irradiated pollen are the number of fruits set and the ratio of parthenogenetic embryos per fruit. Fruit set is lower when

pollination is performed using irradiated pollen [3]. The number of female flowers that develop into a fruit can range between 10–25% in melon [3], 20–25% in pumpkin [8] or 50% in cucumber [9]. The initial number of parthenogenetic embryos is crucial to have enough plant material during the whole process. The plant material usually decreases progressively during each step of the DH process, due to mortality, inefficiency of the method and recalcitrant performance. Frequently, the ratio of parthenogenetic embryos per fruit is low, 0.23–5.79 in cucumber [9,10]; 0.2–16 in pumpkin, squash and winter squash [8,11–14]; 0.3–6 in melon [3,6]; or 1.4 in watermelon [15].

In addition, the genotype of the donor and the receptor plants have an influence in the fruit set. For instance, inbred lines of cucumber resulted in a higher number of parthenogenetic embryos than hybrid lines [16]. On the other hand, the growing environment is another key element to take into account when pollinating with irradiated pollen. During summer/spring the fruit set and the number of embryos is usually higher than in winter/autumn [3,17].

## 2.2. Embryo Detection and Rescue

The use of irradiated pollen to pollinate allows the production of fruits potentially containing parthenogenetic embryos in some of their seeds. However, the vast majority of seeds are empty [3]. Therefore, before embryo rescue, embryo have to be detected to be excised from the seed and cultured in vitro. Three different methods can be used to detect and rescue the parthenogenetic embryos: inspection of seeds one-by-one, X-ray photography and culture of seeds in liquid medium. Each one differs in the time invested, the efficiency and the required equipment (Table 1). The inspection of seeds one-by-one with the help of a binocular microscope is the most widely-used method because it does not require specialized equipment and successfully detects parthenogenetic embryos. Moreover, a light box can be used to ease the inspection of seeds [3,18]. Nevertheless, the inspection of seeds one-by-one is very laborious and time-consuming. On the other hand, the X-ray radiography is the most straightforward method due to is much faster than the inspection of seeds, but requires specialized equipment which is not always available in all laboratories [10]. Lastly, the culture of seeds in liquid medium has been frequently shown to fail, due to contamination with endophytic bacteria and fungi [3,5].

**Table 1.** Summary of the parthenogenetic methodology and efficiency in cucurbit species for doubled haploid (DH) line production.

| Species | Embryo Detection Method | Embryos per Fruit | Mortality Rate In Vitro | Ploidy-Level | Chromosome Doubling | | Reference |
|---|---|---|---|---|---|---|---|
| | | | | | Method | Efficiency | |
| Cucumber | X-ray | 0.23 | 68.23% | 62% H 38% M | E20H8 medium suppl. 500 µM colchicine for 48 h | 55% M 30% DH | Claveria et al. (2005) [10] |
| Cucumber | One-by-one | 5.79 | 79.73% | - | - | - | Smiech et al. (2008) [9] |
| Melon | One-by-one | 6.27 | 30.85% | 73% H 27% M | In vitro solution 500 mg·L$^{-1}$ colchicine for 3 h | 26% DH | Lim and Earle (2008) [7] |
| Melon | X-ray | 0.30 | 50.94% | 73.1 H 23.1% DH 3.8% M | In vivo solution 5000 mg·L$^{-1}$ colchicine for 2 h | 9.38% M 20.31% DH | Hooghvorst et al. (2020) [3] |
| Melon | One-by-one | 1.97 | 34.22% | 90% H | E20H8 medium suppl. 500 µM colchicine for 48 h | - | Gonzalo et al. (2011) [19] |
| Melon | - | - | - | - | In vivo solution 0.5% colchicine for 2 h | 46.03% DH | Solmaz et al. (2011) [20] |
| Pumpkin | One-by-one | 16.38 | 84.04% | 5.9% H | - | - | Kurtar et al. (2009) [13] |
| Pumpkin | One-by-one | 69.85 | - | 0.86% H | | | Košmrlj et al. (2013) [8] |
| Squash | One-by-one | 0.2–10.5 | 71.2–80.1% | 43.7% H 56.3% D | - | - | Kurtar et al. (2002) [12] |
| Squash | One-by-one | 18.45 | - | 65.6% H | | | Baktemur et al. (2014) [14] |
| Watermelon | One-by-one | 1.40 | - | - | - | - | Taskin et al. (2013) [15] |
| Winter squash | One-by-one | 13.72 | 85% | 10.9% H | - | - | Kurtar and Balkaya (2010) [11] |

*H* haploid, *M* mixoploid, and *DH* doubled haploid.

*2.3. In Vitro Culture*

The detected parthenogenetic embryos are rescued and cultured in vitro in specific media. Several media can be used to culture embryos in vitro successfully such as E20A medium [21], MS [22], N6 [23] and CP [24]. Nevertheless, E20A medium with or without modifications is the most commonly used medium for parthenogenetic embryo culture. The parthenogenetic haploid embryos have shorter and irregular cotyledons in comparison to diploid embryos. In addition, they can present a range of morphogenic shapes and stages (pointed, globular, arrow-tip, torpedo, heart, cotyledon, amorphous or necrotic). The survival, germination and development of parthenogenetic embryos is usually correlated with the shape and if its white or necrotic [11]. In addition, during the in vitro process there is a high selection pressure because of deleterious gene combination in homozygosis that can be responsible of vegetative growth problems and can hamper the germination and plantlet development [6,7]. Then, the germination of embryos and the growth and development of the parthenogenetic plantlets is not always guaranteed. The mortality rate of embryos and plantlets in vitro is dramatically high, 30–85% (Table 1).

*2.4. Chromosome Doubling*

Plantlets that survive parthenogenesis are usually haploid (~70%), but mixoploid or even spontaneous DHs can be obtained too [3,5]. Spontaneous DH plantlets do not need to undergo chromosome doubling and can be directly acclimatized for DH seed recovery. On the other hand, haploid and usually mixoploid plantlets have to be chromosome doubled with antimitotic compounds. In cucurbit species, chromosome doubling has been reported chiefly in melon and cucumber. The doubling treatment can be in vitro or in vivo. The antimitotic compound can be colchicine, oryzalin or trifluralin. However, it is mostly performed using colchicine. Dinitroanilines, oryzalin and trifluralin, have been reported as successful and very promising in cucumber [25] and unsuccessful in melon 'Piel de Sapo' [3]. A high mortality might be recorded after the treatment, due to the toxicity of colchicine or the hyperhidricity suffered by the explants. On the other hand, in vivo treatment applies higher concentration of colchicine for a shorter time (2–3 h) by immersing the apical meristems of plants growing in the greenhouse into the colchicine solution. Chromosome doubling is highly influenced by the genotype and the most suitable method must be determined empirically. In melon, the efficiency of chromosome doubling in "Piel de Sapo" type genotype was higher when applying colchicine at 5000 mg·L$^{-1}$ for 2 h in vivo rather than at 500 mg·L$^{-1}$ for 12 h in vitro [3], or in a BC4F1 population, the 26% of haploids chromosome doubled when exposing shoot tip explants to 500 mg·L$^{-1}$ for 3 h in vitro [7]; in cucumber, colchicine was applied in vitro in solid E20H8 medium for 48 h or by submerging plantlet nodes and tips into a colchicine solution for 3–12 h reaching a 55% of chromosome doubling [10] (Table 1).

*2.5. DH Seed Recovery*

Pure and viable seed must be recovered from chromosome doubled and the spontaneous DHs plants. Those plants present a low fruit set (~3–10%) when are self-crossed due to a low germination ability of pollen and the presence of different ploidy-levels in the whole plant [26]. Abnormal ploidy-level in the same plant can be observed in the female and male flowers during the process. The self-crossing must be by hand-pollination to avoid external pollinations. In addition, the germination of DH seed is difficult and a generation for seed multiplication is usually required prior to F1 hybrid seed production.

## 3. Genome Editing in Cucurbit Species

Genome editing techniques have the ability to introduce mutations in the plant genome. There are three main site-specific genome-editing nucleases with the capacity to target precise regions of the genome: zinc finger nucleases (ZFN), transcription activator-like effector nucleases

(TALENs) and clustered regularly interspaced short palindromic repeats associated to nuclease Cas9 (CRISPR/Cas9). The CRISPR/Cas9 system has risen as the preeminent genome editing technique, due to its versatility, efficiency and ease to engineering in comparison to ZFN and TALENs. In cucurbits, there are no reports describing the application of ZNF nor TALENS, as far as the authors know. Nevertheless, CRISPR/Cas9 has been applied successfully in cucumber [27,28], watermelon [29–31] and melon [32].

For the success of a CRISPR/Cas9 experiment, breeders need the sequenced genome of the target specie available, an adequate *Agrobacterium*-mediated transformation protocol and an efficient binary vector containing the sequence of the Cas9 protein capable to induce target mutations. In cucurbit species, the genomic sequences of cucumber, watermelon, melon, pumpkin, zucchini, squash, winter squash, bottle gourd and others are available [33]. Several CRISPR/Cas9 binary vectors have been reported successful in cucumber, watermelon and melon, and may be used for other cucurbits (Table 1). However, *Agrobacterium*-transformation protocol is still the main bottleneck to apply genome editing in cucurbits [31,32].

The production of genome-edited plants in cucurbit species usually starts with the selection of the gRNAs targeting the gene of interest and the construction of the CRISPR/Cas9 binary vector. The CRISPR/Cas9 vector containing the gRNAs should be tested in protoplast prior transformation in order to corroborate the gene editing. The verification of the mutation induction before the *Agrobacterium*-mediated transformation process may ensure its success.

## 3.1. Agrobacterium-Mediated Transformation

The regeneration of transgenic plants in cucurbit species is considered a very recalcitrant process [29,32,34]. To successfully obtain of transgenic plants, regeneration and transformation have to take place. In cucurbits, cotyledonary explants are the main source of plant material for direct organogenesis, which is the production of adventitious buds or shoots from explants without a callus phase. Organogenesis is usually high in cucurbit species when using a suitable regenerating medium. Nevertheless, transformation, the acquisition of the transgene by the germinative cells, is highly inefficient. Therefore, during the process of *Agrobacterium*-mediated transformation, many plants regenerate but the vast majority lacks the transgene. The non-transformed regenerants (so-called "escapes") frequently grow into selective medium, contrary to what occurs in other species. The percentage of "escapes" can be ~30% and are a hindrance in order to select positive transformants. Moreover, the transformation process is usually optimized for and restricted to a few genotypes. This impedes the application of transformation and genome editing to a wide range of genotypes of interest. In comparison to the transformation efficiency of other species such as rice, *Arabidopsis* or tomato, cucurbit species have a low transformation of 1.32–5.63% (Table 2).

**Table 2.** Summary of the methodology and efficiencies of CRISPR/Cas9 reported experiments in cucurbit species.

| Species | Target Gene | Plant Material | Transformation | | Genome Editing Efficiency | Reference |
|---|---|---|---|---|---|---|
| | | | Method | Efficiency | | |
| Cucumber | eIF4E | Cotyledonary explants | *Agrobacterium*-mediated | - | 20% | Chandrasekaran et al. (2016) [28] |
| Cucumber | WIP1 | Cotyledonary explants | *Agrobacterium*-mediated | 1.32% | 65.2% | Hu et al. (2017) [27] |
| Watermelon | PDS | Cotyledonary explants | *Agrobacterium*-mediated | 1.67% | 100% | Tian et al. (2017) [29] |
| Watermelon | PDS | Protoplast | Protoplast transfection | - | 42.1–51.6% | Tian et al. (2017) [29] |
| Watermelon | ACS | Cotyledonary explants | *Agrobacterium*-mediated | - | 23% | Tian et al. (2018) [30] |
| Watermelon | PSK1 | Cotyledonary explants | | 2.3% | - | Zhang et al. (2020) [31] |
| Melon | PDS | Cotyledonary explants | *Agrobacterium*-mediated | 5.63% | 42–45% | Hooghvorst et al. (2019) [32] |
| Melon | PDS | Protoplast | Protoplast transfection | - | 25% | Hooghvorst et al. (2019) [32] |

In addition, the ploidy-level of T0 generation can be spontaneously duplicated during the direct organogenesis resulting in tetraploid regenerants. Those polyploid plants are sterile and selfing is impossible for segregation of the transgene or for transgenic seed multiplication.



*3.2. Genome Editing Efficiency*

The mutation efficiency depends on the GC content of the sgRNA, the number of transformed cells and the Cas9 protein expression level in transgenic cells [35]. In species such as rice or Arabidopsis, the efficiency frequently ranges between 80–100% and in cucurbit species it ranges from 20–100% (Table 2). Taking into account the larger effort invested in rice or Arabidopsis to apply and optimize CRISPR/Cas9 in comparison to cucurbit species, CRISPR/Cas9 efficiency in cucurbits can be recognized as acceptable and suitable. Then, genome editing efficiency is not a bottleneck when applying CRISPR/Cas9 in cucurbits. Nevertheless, further attempts should be assayed, as done in other species, to achieve a higher genome editing efficiency such as using endogenous promoters of Cas9 and sgRNA expression, heat treatment during transformation, optimization of transformation efficiency [36,37].

## 4. Haploid Inducer-Mediated Genome-Editing in Cucurbit Species

Haploid inducer approach is based on an intraspecific cross between a haploid inducer line and a receptor genotype of interest from where to obtain haploid lines. The haploid inducer line carries a specific mutation in an essential gene for the normal fertilization of female cells and therefore induces the parthenogenetic development of haploid embryo from the egg cell. The MATRILINEAL (MATL) gene, also known as NOT LIKE DAD (NLD) or PHOSPHOLIPASE A1 (PLA1) encodes a pollen-specific phospholipase and is usually the mutated gene that causes the haploid inducer ability in cereals [38–40]. For instance, in maize, natural haploid inducer lines were discovered to carry a 4-bp insertion in the carboxy terminus of the MATL gene. Natural haploid inducer lines have been used in wheat, maize, tobacco or barley for years to induce a parthenogenetic process for further haploid line obtention [40–44]. In cucurbit species, no natural haploid inducer lines have been described, as far as the authors know. However, with the recent genome editing tools available, breeders have the possibility to generate haploid inducer lines through genome-editing techniques. Haploid inducer lines have been produced in cereal species mutating MATL gene via CRISPR/Cas9. Haploid inducer-mediated genome-editing approach yielded a 6.7% of parthenogenetic haploids in maize [45] and ~6% in rice [46]. Those rates represent a great improvement in in vivo doubled haploid obtainment in those species to allow the acceleration of breeding.

The MATL gene is present and highly conserved in cereal species but not in dicots. Therefore, the haploid inducer-mediated genome-editing system in dicots usually targets centromeric histone 3 (CENH3) gene. The CENH3 gene codes a histone present in all plants that determines the position of the centromere, and, thus, plays a major role in chromosome segregation during mitosis [47]. The haploid inducer related potential of CENH3 gene was first discovered in *Arabidopsis*. In this study, a haploid inducer line named "green fluorescent protein (GFP)-tailswap" carried GFP fused to the N-terminal tail domain of an H3 variant, replacing the N-terminal tail of CENH3. This CENH3 mutant line produced a haploid induction rate of 25–45% when crossed with a CENH3 wild type line by chromosomes elimination of the mutant line (Table 3) [48]. Since the evidence of the potential of CENH3 for haploid induction, some applications have raised using ethyl methane sulfonate (EMS) mutagenesis targeting CENH3 gene to obtain haploid inducing lines, such as in tomato with a haploid progeny of 0.2–2.3% (WO 2017 200386/KEYGENE); in cucumber, with an efficiency of 1%; and in melon, with an efficiency 1.5% (WO 2017 081,011 A1/RIJK ZWAAN) (Table 3). However, those procedures are patented, and EMS mutagenesis is quite inefficient in those species.

Therefore, the generation of haploid inducer lines in cucurbit species may be democratized targeting CENH3 using CRISPR/Cas9, due to the specificity of CRISPR/Cas9 in multiple sequences of the CENH3 gene which can be targeted efficiently. However, for the generation of haploid inducer lines mutated via CRISPR/Cas9 a successful transformation protocol would be imperative at least for one genotype. Generally, transformation protocol is adjusted for a specific genotype in cucurbit species. Fortunately, the genotype of the haploid inducer line is not an issue for haploid induction as long as it is the same species as the donor genotype from where to recover haploid embryos and bears male flowers. From the T0 CRISPR/Cas9 generation, heterozygous CENH3-mutated should be self-crossed in order

to select: (i) T1 transgene-free and homozygous CENH3-mutated individuals, and (ii) transgene-free and heterozygous mutated-CENH3 T1 individuals. The homozygous CENH3-mutated lines will be used for haploid induction process and heterozygous lines for haploid inducer line maintenance for future applications. Through this method, haploid inducer lines may be successfully maintained and used over the time (Figure 1).

**Table 3.** Haploid induction reports mediated by mutation or disruption of centromeric histone 3 (CENH3), MATRILINEAL (MATL) or DMP.

| Species | Target Gene | Mutation or Disruption Method | Haploid Progeny | Reference |
|---|---|---|---|---|
| Arabidopsis | CENH3 | GFP-tailswap disruption | 4–34% | Ravi and Chan (2010) [48] |
| Arabidopsis | CENH3 | GFP-tailswap disruption | 2.50% | Kuppu et al. (2015) [49] |
| Arabidopsis | CENH3 | GFP-tailswap disruption | 4.80% | Karimi-Ashtiyani et al. (2015) [50] |
| Arabidopsis | CENH3 | GFP-tailswap disruption | 3.60% | Kelliher et al. (2016) [51] |
| Arabidopsis | DMP | CRISPR/Cas9 | 2.1% | Zhong et al. (2020) [52] |
| Cucumber | CENH3 | EMS-induced mutation | 1% | WO 2017 081,011 A1/RIJK ZWAAN |
| Maize | CENH3 | GFP-tailswap disruption | 3.6% | Kelliher et al. (2016) [51] |
| Maize | MATL | TALEN | 6.70% | Kelliher et al. (2017) [45] |
| Maize | DMP | CRISPR/Cas9 | 0.1–0.3% | Zhong et al. (2019) [53] |
| Melon | CENH3 | EMS-induced mutation | 1.50% | WO 2017 081,011 A1/RIJK ZWAAN |
| Rice | CENH3 | EMS-induced mutation | 1% | WO 2017 200386/KEYGENE |
| Rice | MATL | CRISPR/Cas9 | 2–6% | Yao et al. (2018) [46] |
| Tomato | CENH3 | GFP-tailswap disruption | 0.2–2.3% | WO 2017 200386/KEYGENE |
| Wheat | MATL | CRISPR/Cas9 | 18.9% | Liu et al. (2019) [54] |

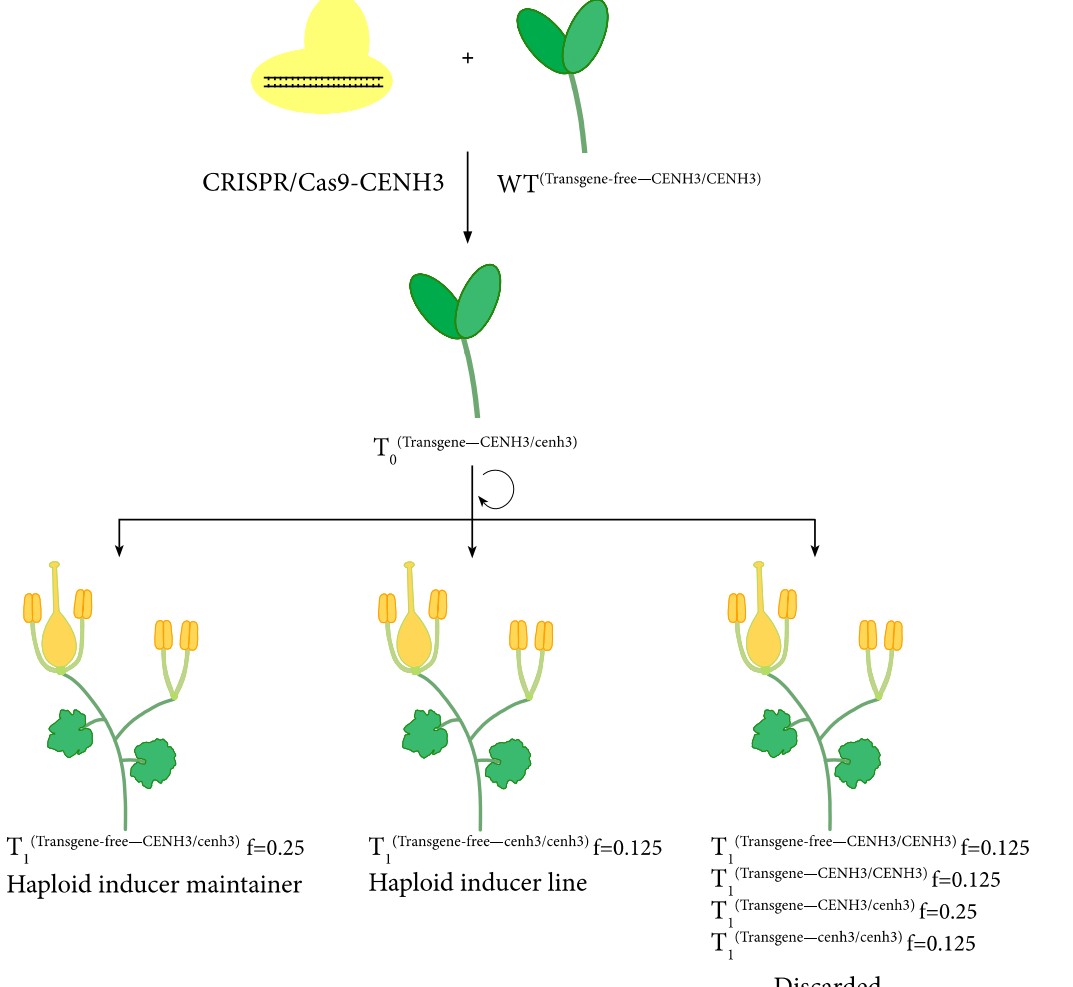

**Figure 1.** Schematic representation of the obtention of a haploid inducer line and its maintainer. WT: wild type.

The application of the haploid inducer-mediated genome-editing system in cucurbit species could reduce the actual time-consuming, labor-intensive and limited parthenogenic process. Although the process would remain very similar to parthenogenetic process with irradiated pollen due to the initial in vivo pollination with pollen of the haploid inducer line, line, the rate of haploid embryo induction could be significantly improved since no aggressive treatment, such as ionization, would be applied in haploid inducer line pollen. Furthermore, no irradiation source requiring equipment not always available in all laboratories will be needed. On the other hand, parthenogenesis via irradiated pollen is highly genotype-dependent and accomplishment of irradiation depends upon the radiation resistance of the pollen. In addition, some genotypes are reported to be very recalcitrant to induction of haploid generations when pollinated with irradiated pollen. Pollination with pollen of the haploid inducer lines could broaden the range of genotypes that can produce haploids and increase the number of parthenogenetic embryos. This must be assayed to see whether haploid inducer lines can increase the number of haploid embryos produced in cucurbits.

Moreover, parthenogenesis with irradiated pollen is routinely applied in cucumber and melon and substantial progress has been made. In contrast, less effort has been applied in species such as watermelon, winter squash, pumpkin or bottle gourd and, therefore, less progress has been made in optimizing their parthenogenetic protocols. In cucurbit species, the fruits set once pollinated with irradiated pollen or pollen from haploid inducer lines will follow the same steps as in in situ parthenogenesis via irradiated pollen. Consequently, limiting factors described for parthenogenesis with irradiated pollen would likewise be present using the haploid inducer approach. Therefore, the haploid inducer-mediated genome-editing approach can take advantage of the progress made to successfully obtain doubled haploid lines.

The haploid inducer-mediated genome-editing approach is an opportunity to improve the efficiency of doubled haploid production in recalcitrant species. The use of matl or cenh3 mutant lines in cereals or dicots for haploid induction and production might avoid androgenesis, gynogenesis or parthenogenesis via pollen irradiated and boost their obtention in several species.

## 5. Regulatory Landscape for the New Generation of Doubled Haploids

The development of DH lines has never been restricted by regulatory limitation, because they are obtained using in vitro protocols for haploid generation or in vivo pollination with pollen mutagenized with gamma rays. On the other hand, since the Directive 2001/18/EC put in place in EU, GMOs have been strongly limited because of uncertainty and safety. The GMO definition is "as an organism in which the genetic material has been altered in a way that does not occur naturally by mating and/or natural recombination". The European highest court handed a setback regulation on 25 July of 2018, subjecting organisms obtained using CRISPR/Cas9 and other genome editing techniques under the same regulation as GMOs, defining them as "recombinant nucleic acid techniques involving the formation of new combinations of genetic material". Therefore, GMOs and CRISPR/Cas9-derived plants have the same regulation landscape despite the fact that the first added a transgene essential for the trait improvement whereas the second used recombinant DNA for trait improvement and subsequently lost it through segregation. Then, as pointed out by Abbot, the EU criterion is based on the process and the product when using recombinant DNA [55].

Doubled haploids derived from haploid inducer-mediated genome-editing approach presents a challenge in terms of regulation. When haploid lines are produced by crossing the haploid inducer line with a receptor genotype, the chromosomes of the haploid inducer line are eliminated, and the haploid-derived progeny carries the maternal set of chromosomes. The only progenitor of the haploid generation has never carried recombinant DNA and has no improved traits derived from mutations induced by the genome editing technique. The origins of haploid lines derived from haploid inducer or irradiated pollen are impossible to trace. Therefore, haploid-mediated genome-editing system should not be restricted by any limitation or regulation.

**Author Contributions:** I.H.: Conceptualization, writing, review and editing. S.N.: Supervision, project administration and funding acquisition. Both authors have read and agreed to the published version of the manuscript. All authors have read and agreed to the published version of the manuscript.

**Funding:** This research was funded by Departament d'Innovació, Universitats i Empresa, Generalitat de Catalunya, 2017 DI 001.

**Conflicts of Interest:** The authors declare no conflict of interest.

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
