# Peer review of "Opportunities and Challenges in Doubled Haploids and Haploid Inducer-Mediated Genome-Editing Systems in Cucurbits"

_agronomy, doi:10.3390/agronomy10091441_

Round 1
Reviewer 1 Report
All reviewer's remarks are in the eclosed file.

Author Response
Reviewer 1
The authors have improved the quality of the analyzed data. However, there still are inaccuracies that must be corrected.
- The table 1 has been well improved, but there are some inaccuracies that have not been still corrected.
Košmrlj et al. (2013) [7] As given 0.86% H – «plody level» in the manuscript, but in the cited publication it is given as «…Of 1397 plantlets analyzed, 12 were haploid, 1376 diploid, one triploid, and eight were Tetraploid» , here, it must be added as: 98.5 % - diploid, 0.01% - triploid, 0.06% - Tetraploid
Baktemur et al. (2014) [13] it is given as 0.66 % H – «plody level» in the table of the manuscript, but in the publication of Baktemur et al. (2014) [13] it goes as «…8 genotypes and the same gamma doses were used, and 2625 haploid and 1378 diploid embryos were obtained from 217 fruit..» If it is counted well it would be as: 65,6% haploid, 34,4% diploid embryos
The authors should thoughtfully verify other data presented in the table 1, since it is not a task of the reviewer to analyze data.
Author: In the manuscript of Košmrlj et al. (2013), they assay several gamma doses (0, 50, 100, 150, 200, 300, and 350 Gy), and the total sum of 1397 are plants derived from those doses. The authors of the paper do not specify if the 1378 diploid plants are spontaneous doubled haploid or diploid. We feel that those diploid plants are not doubled haploids since they represent almost the 100% of obtained plants.
Indeed, we made a mistake, instead of 0.66% it was 65.6%. We have changed it.
Our Table 1 do not attempt to depict all the data exposed in the papers we cite. We just want to express the minimum amount of data that will be useful for those attempting to have an overview of the technique and its efficiencies. Since diploid, triploid and tetraploids plants are not useful ploidy-levels in squash for breeding purposes, we preferred to not add it.
- Line 217 - 220 “The MATRILINEAL (MATL) gene, also known as NOT LIKE DAD (NLD) or PHOSPHORYLASE A1 (PLA1) encodes a pollen- specific phospholipase and is usually the mutated gene that causes the haploid inducer ability in cereals [30–32]. – There is no information on the subject in these articles cited in the manuscript. Consequently, these references are not correct.
Author: Indeed, those references are not according to the subject. We had an issue concerning the references numeration and all of them were moved. We checked one by one carefully but this one may had been disregarded.
- Figure 1. The abbreviation should be given. Is WT a wild type? The allele composition of WT should be also shown. There is no notification either male flowers or hermaphroditic flowers in the text under the schema. It is right? Because it is not quite clear the flowers must be either male/female or bisexual/hermaphroditic ones.
The abbreviation of WT and its allele composition have been added.
There is no notification in the flower phenotype because it does not change after the CENH3 mutation. The pollen viability does.
Also it is not correctly named first genotype T1 among discarded plants. It is shown same as in T1 haploid inducer maintainer, but probably it is very likely to be T1 transgene-free CENH3/CENH3.
We have changed the issue.
Reviewer 2 Report
The article is addressing interesting topic and well presented. I have some major concerns as listed below which authors can addressed while revising their MS.
Title needs improvement – at present it says haploid mediated genome-editing which should be reversed “genome-editing mediated haploid induction in cucurbits”
Use of haploid and double haploid term dose not seems ok, authors can keep either of it.
Abstract
Abstract is to small and does not provide snapshot of the review. I guess author can add couple of sentence enlisting discussed approaches.
Introduction
Authors needs to add information to explain how exactly double haploid development help in cucurbit breeding program. At least one paragraph should be added.
“Pure seed must be recovered from chromosome doubled” - is this “Pure” or “viable seed”
The subsection “Agrobacterium-mediated transformation” does not fit well in order as well as need to elaborate further
Authors have missed several important efforts with cucurbit genome editing, you can get such information from recent review published in Cells by Vats et al. 2020.
In table 2, the column “Genome editing technique” can be removed since all the examples used have the same techniques. The information can be added in the table title.
Please confirm how to denote “CENH3” or “CenH3”
Figure 1 need significant improvement. Please add segregation percentages to get better idea.
Is it possible to add conclusion section.
Author Response
Reviewer 2
The article is addressing interesting topic and well presented. I have some major concerns as listed below which authors can addressed while revising their MS.
Title needs improvement – at present it says haploid mediated genome-editing which should be reversed “genome-editing mediated haploid induction in cucurbits”
Author: the Haploid-Inducer Mediated Genome Editing (IMGE) approach is usually referred as it is. We do not feel the need to change. Please check the following articles as examples:
- Development of a Haploid-Inducer Mediated Genome Editing System for Accelerating Maize Breeding
- Extension of the in vivo haploid induction system from diploid maize to hexaploid wheat
Use of haploid and double haploid term dose not seems ok, authors can keep either of it.
Author: We disagree with the reviewer, as the doubled haploid technique in cucurbits it is usually referred to the process of pollinate with irradiated pollen until the obtainment of doubled haploids by chromosome doubling. However, the haploid-inducer technique specifically refers to the fact of obtaining haploid from an haploid inducer line. We understand the reviewer’s point but we don’t feel the need of changing the main title of the MS.
Abstract
Abstract is to small and does not provide snapshot of the review. I guess author can add couple of sentence enlisting discussed approaches.
Author: we have added a couple of sentences as the reviewer recommended.
Introduction
Authors needs to add information to explain how exactly double haploid development help in cucurbit breeding program. At least one paragraph should be added.
Author: We have added a paragraph of the use of DH in breeding as recommended by the reviewer.
“Pure seed must be recovered from chromosome doubled” - is this “Pure” or “viable seed”
Author: both concepts are applicable. We have added the consideration of viable.
Authors have missed several important efforts with cucurbit genome editing, you can get such information from recent review published in Cells by Vats et al. 2020.
Author: Despite of the excellence of the recommended article, we do not see the important efforts in cucurbit genome editing missed in our manuscript.
In table 2, the column “Genome editing technique” can be removed since all the examples used have the same techniques. The information can be added in the table title.
Author: we have removed the column.
Please confirm how to denote “CENH3” or “CenH3”.
CENH3 is usually expressed in capital letters.
Figure 1 need significant improvement. Please add segregation percentages to get better idea.
Authors: we have added the frequencies.
Is it possible to add conclusion section.
Author: We have added a brief conclusion in each subsection. We don’t feel the need of a general conclusions section, as this MS does not explain new basic research. It just give insights on the research done and the usefull future research in DH cucurbit breeding.
Reviewer 3 Report
Review of Agronomy-898441
Opportunities and Challenges in Doubled Haploids and Haploid Inducer-Mediated Genome-Editing Systems in Cucurbits
Isidre Hooghvorst, Salvador Nogués
General comments:
Overall, this is a useful review that provides a lot of information on ways to create doubled haploids for cucurbit breeding and on the application of CRISPR/Cas9 technologies in cucurbits, especially with regards to the potential use of this technology to expedite the production of doubled haploids. The authors have done a thorough job of first explaining why it is useful for cucurbit breeding and then describing the current state of the field and provide numerous examples of ways that doubled- haploids have been created. Table I listing specific examples of how doubled-haploids have been created and detected is very useful. Similarly, Table 2 provides a good summary of the current status of genome editing in cucurbits using CRISPR/Cas9. The authors finish with some useful insights and suggestions for the further development of the field.
The paper need correct the following errors:
Lines 14- 24: Now they have left out the last 3 lines of the abstract.
The abstract should be : ” In situ parthenogenesis via irradiated pollen is the preferred technique to obtain haploid plantlets whose chromosomes are then doubled in Cucurbitaceae such as melon, cucumber, pumpkin, squash and winter squash. In contrast to doubled haploid procedures in other species, in situ parthenogenesis in cucurbits presents many limiting factors which impede efficient production of haploids. In addition, it is very time-consuming and labor-intensive. However, the haploid inducer-mediated genome-editing system is a breakthrough technology for producing doubled haploids. Several reports have described using the CRISPR/Cas9 system in cucurbit species and although its application has many bottlenecks the targeted knock-out of the CENH3 gene will allow breeders to obtain haploid inducer lines that can be used to obtain parthenogenetic embryos. In this review, we discuss the progress made towards the development of doubled haploids and haploid inducer genotypes using CRISPR/Cas9 technologies in cucurbit species. The present review provides insights for the application of haploid inducer-mediated genome-editing system in cucurbit species.”
Line 33 should be: “…due to heterosis that results in earlier harvest, increased yield and higher vigor [2].”
Line 49 should be: “Until recently, classical breeding…”
Lines 65-68 should be: “The parthenogenesis process starts with in vivo pollination using irradiated pollen. The resulting parthenogenetic embryos are then detected, rescued, and cultured in vitro in order to germinate and develop into plantlets. “
Line 78 should be: “…and grow…”
Line 79 should be: “However, this pollen is …”
Lines 85 should be: “The parameters that define the success or failure of pollination with irradiated pollen are the number…”
Lines 106 - 107 should be: “…help of a binocular microscope is the most widely-used method because it does not require specialized equipment and successfully detects parthenogenetic embryos. “
Line 112 should be: “…endophytic bacteria and fungi.”
Lines 115-119 should be: “The detected parthenogenetic embryos are rescued and cultured in vitro in specific media. Several media can be used to culture embryos in vitro successfully such as E20A [18], MS 116 [19], N6 [20] and CP [21]. Nevertheless, E20A medium with or without modifications is the most commonly-used medium for parthenogenetic embryo culture. The parthenogenetic haploid embryos usually have shorter and irregular cotyledons in comparison to diploid embryos.”
Line 124 should be: “…gene combinations in homozygotes that can be responsible for vegetative growth problems”…
Line 164 should be: “…efficiency and ease of engineering in comparison…”
Line 171 should be: “…the genomic sequences of cucumber …”
Line 178 should be: “…should be tested in protoplasts prior to transformation …”
Lines 182-183 should be: The regeneration of transgenic plants in cucurbit species is considered a very recalcitrant process [30], [33], [35]. To successfully obtain transgenic plants…”
Line 258 should be: Figure 1. Schematic representation of the obtention of a haploid inducer line and its maintainer.
Lines 259-262 should be: “The application of the haploid inducer-mediated genome-editing system in cucurbit species could reduce the actual time-consuming, labor-intensive and limited parthenogenic process. Although the process would remain very similar to parthenogenetic process with irradiated pollen due to the initial in vivo pollination with pollen of the haploid inducer line, line, the rate of haploid embryo induction could be significantly…”
Lines 264-271 should be: “Furthermore, no irradiation source requiring equipment not always available in all laboratories will be needed. On the other hand, parthenogenesis via irradiated pollen is highly genotype-dependent and accomplishment of irradiation depends upon the radiation resistance of the pollen. Besides, some genotypes are reported to be very recalcitrant to induction of haploid generations when pollinated with irradiated pollen. Pollination with pollen of the haploid inducer lines could broaden the range of genotypes that can produce haploids and increase the number of parthenogenetic embryos. This must be assayed to see whether haploid inducer lines can increase the number of haploid embryos produced in cucurbits.”
Lines 278-279 should be: “…would likewise be present using the haploid inducer approach. Therefore, the haploid …”
Line 284 should be: “…via irradiated pollen and…”
Lines 296-298 should be: “…same regulation landscape despite the fact that the first added a transgene essential for the trait improvement whereas the second used recombinant DNA for trait improvement and subsequently lost it through segregation. Then, as pointed out by Abbot, …”
Lines 306-307 should be: “Therefore, the haploid-mediated genome-editing system should not be restricted by any limitation or regulation.”
Author Response
We have incorporated all the recommendations made by the reviewer. We kindly appreciate the effort of the reviewer in the correction of English lenguage.
Round 2
Reviewer 2 Report
Title needs improvement – at present it says haploid mediated genome-editing which should be reversed “genome-editing mediated haploid induction in cucurbits”
Author: the Haploid-Inducer Mediated Genome Editing (IMGE) approach is usually referred as it is. We do not feel the need to change. Please check the following articles as examples:
- Development of a Haploid-Inducer Mediated Genome Editing System for Accelerating Maize Breeding
The article in maiz used a haploid inducer to eliminate the CRISPR/Cas9 since the paternal chromosome get eliminated (which has CRISPR/Cas9 cassette) but the edited gene on the maternal chromosome will pass to the next progeny. The same principle used for another paper with wheat.
If authors prefer to go with the present title, this is their article to decide.
This manuscript is a resubmission of an earlier submission. The following is a list of the peer review reports and author responses from that submission.
Round 1
Reviewer 1 Report
All information in file inclosed.

Author Response
The authors significantly improved the quality of the analyzed data. However still needs few improvements.
Main suggestions:
1. The name of the family should be given in normal font. For the bigger taxon the normal font is usually used. As it goes “The Cucurbitaceae”.
We have changed the font.
- The table 1 has been well improved, but there are few moments to be given more precisely:
It is not quite clear how the authors calculated such characteristics as
“Embryos per fruit” and “Ploidy level”, since there is no information on the subject in cited article
Here, it should be given either the exact reference or the method of calculation.
For example:
Košmrlj et al. (2013) [7]
To explain:
153.20 – «embryos per fruit»it is given in the current manuscript, although in the cited publication of Košmrlj et al. (2013) [7] it is described as «In a control treatment (pollination with non-irradiated pollen), fruit set and mean number of embryos per 100 seeds were 55.6% and 73.4, respectively. A gradual decrease of both parameters was observed at increased irradiation doses. At the highest irradiation dose (350 Gy), fruit set and mean number of embryos was 25.0% and 18.8, respectively...»
We wrongly assumed that the number of Flowers pollinated, and the % of fruit set was for experiments from 2011 ad 2012. We made a huge mistake.
As given 0.39% H – «plody level»in the manuscript, but in the cited publication it goes as «The observed percentages of haploid embryos ranged from 0% in ‘Rumena golica’ and ‘Naked Seed’ to 10% in ‘Turkey #2’ for pollen irradiated at 200 Gy and from 0% in ‘Slovenska golica’ to 3.9% in ‘Naked Seed’ for pollen irradiated at 300 Gy (Table 4) ...». Also as in reference – «...Of 1397 plantlets analyzed, 12 were haploid, 1376 diploid, one triploid, and eight were Tetraploid»
Same wrong assumption was done for ploidy level.
We have changed the values and corrected it. The values are just for the 2011 experiment, expressed in the table 2 and 3 at Kosmij et al 2013. We count 20 fruits and 1397 embryos. Ploidy level, is expressed according to the statement “1397 plantlets analyzed, 12 were haploid, 1376 diploid, one triploid, and eight were Tetraploid”
Baktemur et al. (2014) [13]
it is given as 100 % H – «plody level»in the table of the manuscript, but in the publication of Baktemur et al. (2014) [13] it goes as «...8 genotypes and the same gamma doses were used, and 2625 haploid and 1378 diploid embryos were obtained from 217 fruit..»
We did another mistake as we did not add the number of diploid embryos (1378). Therefore, the number of embryos per fruits was under calculated and the % of haploids over calculated. We have corrected it.
Minor suggestions:
The cited list of publications should be corrected in accordance with requirement to the journal (page indications and so on), there are inaccuracies in 7, 8, 17, 31, 32, 37
We have corrected the references.
Reviewer 2 Report
The review article is interesting. I have few major and minor suggestion as listed below.
Abstract
“CRISPR/Cas9 system have been reported several times in cucurbit species” – need improvement
Subheading need to be more elaborated
“DH lines in cucurbit species are mainly produced parthenogenetically”. – There are many such sentences, starting with the abbreviations, better not to start a sentence with an abbreviation.
Line 109 – “the efficiency and the required equipment. (Table 1)” –typo
Line 176 - Several CRISPR/Cas9 binary vectors have been reported successful in cucumber, watermelon and melon and may be used for other cucurbits. – need citation
Time needed to get genome-edited transgene plants and the issue related to the maintenance of the CenH3 mutant lines need to discuss in details.
Genes other than the CenH3 need to describe in details.
Authors contribution – “All authors ..” there are only two authors.
Need to add summary figure for CenH3 approach. See Vats et al. Cells 2019, 8(11), 1386; https://doi.org/10.3390/cells8111386
Similarly, better to have figure showing all the available approaches for haploid developments in cucurbits.
Author Response
Reviewer 2
The review article is interesting. I have few major and minor suggestion as listed below.
Abstract
“CRISPR/Cas9 system have been reported several times in cucurbit species” – need improvement
We have improved the sentence.
Subheading need to be more elaborated
We have added some specification in some subheading.
“DH lines in cucurbit species are mainly produced parthenogenetically”. – There are many such sentences, starting with the abbreviations, better not to start a sentence with an abbreviation.
We have tried to change the sentences with a starting abbreviation. However, there are some remaining, such as for CRISPR.
Line 109 – “the efficiency and the required equipment. (Table 1)” –typo
We have corrected it.
Line 176 - Several CRISPR/Cas9 binary vectors have been reported successful in cucumber, watermelon and melon and may be used for other cucurbits. – need citation
We have added the Table 1 captation since all reported CRISPR experiments in cucurbits are summarized there.
Time needed to get genome-edited transgene plants and the issue related to the maintenance of the CenH3 mutant lines need to discuss in details.
Genes other than the CenH3 need to describe in details.
Authors contribution – “All authors ..” there are only two authors.
We have corrected it.
Need to add summary figure for CenH3 approach. See Vats et al. Cells 2019, 8(11), 1386; https://doi.org/10.3390/cells8111386
Similarly, better to have figure showing all the available approaches for haploid developments in cucurbits.
We do not feel the need of the proposed figure because the review article by Dong et al. 2016 about DH technology in cucurbits summarize better this aspect of the technique.
Reviewer 3 Report
Review of Agronomy-848961
Opportunities and Challenges in Doubled Haploids and Haploid Inducer-Mediated Genome-Editing Systems in Cucurbits
Isidre Hooghvorst, Salvador Nogués
General comments:
Overall, this is a useful review that provides a lot of information on ways to create doubled haploids for cucurbit breeding and on the application of CRISPR/Cas9 technologies in cucurbits, especially with regards to the potential use of this technology to expedite the production of doubled haploids. The authors have done a thorough job of first explaining why it is useful for cucurbit breeding and then describing the current state of the field and provide numerous examples of ways that doubled- haploids have been created. Table I listing specific examples of how doubled-haploids have been created and detected is very useful. Similarly, Table 2 provides a good summary of the current status of genome editing in cucurbits using CRISPR/Cas9. The authors finish with some useful insights and suggestions for the further development of the field.
Unfortunately, although improved, the English still needs a lot of work.
Here are some specific comments, but there are many more errors that I don’t have time to flag.
Lines 15-23 should be” In situ parthenogenesis via irradiated pollen is the preferred technique to obtain haploid plantlets whose chromosomes are then doubled in Cucurbitaceae such as melon, cucumber, pumpkin, squash and winter squash. In contrast to doubled haploid procedures in other species, in situ parthenogenesis in cucurbits presents many limiting factors which impede efficient production of haploids. In addition, it is very time-consuming and labor-intensive. However, the haploid inducer-mediated genome-editing system is a breakthrough technology for producing doubled haploids. Several reports have described using the CRISPR/Cas9 system in cucurbit species and although its application has many bottlenecks the targeted knock-out of the CENH3 gene will allow breeders to obtain haploid inducer lines that can be used to obtain parthenogenetic embryos. In this review, we discuss…”
Lines 34-36 are difficult to understand and grammatically-incorrect and should be rewritten.
Lines 37-38 should be: “Therefore, pure lines used as parents for hybrid production are invaluable. There are two main methods in cucurbits to produce completely homozygous lines, classical breeding and doubled haploids.”
Lines 46-51 should be: “bottlenecks and limitations that may jeopardize the final production of DH lines such as: high genotypic dependency; low fruit set when pollinated with irradiated pollen; difficulties in detecting parthenogenetic embryos; low production of parthenogenetic embryos; recalcitrant culture in vitro; high mortality of parthenogenetic plants during in vitro culture and after the chromosome doubling treatment; low rates of chromosome doubling; and, low fruit set of DH lines once chromosomes are doubled [3].”
Lines 54-55 should be: “The currently-predominant genome editing technique is CRISPR/Cas9 due to its efficiency, versatility, and unprecedented control over mutation.”
Lines 59-63 should be: “ There are numerous reports and reviews of DH production and CRISPR/Cas9 applications in many plant species including cucurbits. Nevertheless, haploid inducer-mediated genome-editing systems have not yet been applied in cucurbits. Therefore, the purpose of this review is to focus on haploid inducer-mediated genome-editing systems in cucurbit species to give new insights, opportunities and challenges that may be valuable for developing this technique in cucurbits and other species.”
Lines 65-69 are difficult to understand and grammatically-incorrect and should be rewritten.
Lines 74-77 should be: “Once the plantlets have doubled their chromosomes, they are cultured in the greenhouse together with the spontaneously-doubled haploid lines to recover DH seed [3,5,6]. During the DH process, a high genotypic dependency and other factors continuously hamper each step causing a loss of efficiency that might be critical.”
Lines 80-81 please rewrite. Please explain what is “gammacel.” Is this the source of radiation? Should this be Gammacell? Also please change line 81 to “…irradiation sources applied to male flowers”
Lines 81 -88 should be “Although ionized pollen can germinate on the female stigma and grow pollen tubes to reach the embryo sac, they are unable to fertilize the egg-cell and the polar nuclei. Therefore, irradiated pollen stimulate egg-cell division and induce parthenogenetic embryos [3]. Overall, the dose of ionizing radiation can range from 25 to 500 Gy depending on the species and can yield less parthenogenetic haploid embryos at higher or more diploid embryos at lower dosages. Therefore irradiation should be optimized for each species because pollen sensitivity is attributed to radiation-resistance.”
Lines 104-107 are grammatically-incorrect and should be rewritten.
Lines 109-113 are grammatically-incorrect and should be rewritten.
Lines 126-130 are difficult to understand and grammatically-incorrect and should be rewritten.
Lines 141 should be “Colchicine was applied in vitro in solid E20H8 medium for 48h…”
Lines 146-147 should be “Chromosome doubling is highly influenced by the genotype and the most suitable method must be determined empirically.”
Lines 152-154 are difficult to understand and grammatically-incorrect and should be rewritten.
Lines 182-184 require clarification. Do you mean that the vectors need to be tested in protoplasts before they are used for plant transformation? Do you just test for expression of the gRNA and Cas9 protein, or do you test whether the targeted gene is actually edited in protoplasts?
Lines 196-197 should be: Moreover, the transformation process is usually optimized for and restricted to a few genotypes. This impedes the application of transformation and genome editing to a wide range of genotypes of interest.
Lines 255-256 “free-transgene” should be “transgene-free”
Line 267: what does “pollen-radio” mean?
Author Response
Reviewer 3
General comments:
Overall, this is a useful review that provides a lot of information on ways to create doubled haploids for cucurbit breeding and on the application of CRISPR/Cas9 technologies in cucurbits, especially with regards to the potential use of this technology to expedite the production of doubled haploids. The authors have done a thorough job of first explaining why it is useful for cucurbit breeding and then describing the current state of the field and provide numerous examples of ways that doubled- haploids have been created. Table I listing specific examples of how doubled-haploids have been created and detected is very useful. Similarly, Table 2 provides a good summary of the current status of genome editing in cucurbits using CRISPR/Cas9. The authors finish with some useful insights and suggestions for the further development of the field.
Unfortunately, although improved, the English still needs a lot of work.
We would like to thank the reviewer for the English corrections. We think that the recommendations have improves substantially the quality of the language.
Here are some specific comments, but there are many more errors that I don’t have time to flag.
Lines 15-23 should be” In situ parthenogenesis via irradiated pollen is the preferred technique to obtain haploid plantlets whose chromosomes are then doubled in Cucurbitaceae such as melon, cucumber, pumpkin, squash and winter squash. In contrast to doubled haploid procedures in other species, in situ parthenogenesis in cucurbits presents many limiting factors which impede efficient production of haploids. In addition, it is very time-consuming and labor-intensive. However, the haploid inducer-mediated genome-editing system is a breakthrough technology for producing doubled haploids. Several reports have described using the CRISPR/Cas9 system in cucurbit species and although its application has many bottlenecks the targeted knock-out of the CENH3 gene will allow breeders to obtain haploid inducer lines that can be used to obtain parthenogenetic embryos. In this review, we discuss…”
We have added the recommendations.
Lines 34-36 are difficult to understand and grammatically-incorrect and should be rewritten.
We have deleted the sentence because of the lack of importance.
Lines 37-38 should be: “Therefore, pure lines used as parents for hybrid production are invaluable. There are two main methods in cucurbits to produce completely homozygous lines, classical breeding and doubled haploids.”
We have added the recommendations.
Lines 46-51 should be: “bottlenecks and limitations that may jeopardize the final production of DH lines such as: high genotypic dependency; low fruit set when pollinated with irradiated pollen; difficulties in detecting parthenogenetic embryos; low production of parthenogenetic embryos; recalcitrant culture in vitro; high mortality of parthenogenetic plants during in vitro culture and after the chromosome doubling treatment; low rates of chromosome doubling; and, low fruit set of DH lines once chromosomes are doubled [3].”
We have added the recommendations.
Lines 54-55 should be: “The currently-predominant genome editing technique is CRISPR/Cas9 due to its efficiency, versatility, and unprecedented control over mutation.”
We have added the recommendations.
Lines 59-63 should be: “ There are numerous reports and reviews of DH production and CRISPR/Cas9 applications in many plant species including cucurbits. Nevertheless, haploid inducer-mediated genome-editing systems have not yet been applied in cucurbits. Therefore, the purpose of this review is to focus on haploid inducer-mediated genome-editing systems in cucurbit species to give new insights, opportunities and challenges that may be valuable for developing this technique in cucurbits and other species.”
We have added the recommendations.
Lines 65-69 are difficult to understand and grammatically-incorrect and should be rewritten.
We have rephrased.
Lines 74-77 should be: “Once the plantlets have doubled their chromosomes, they are cultured in the greenhouse together with the spontaneously-doubled haploid lines to recover DH seed [3,5,6]. During the DH process, a high genotypic dependency and other factors continuously hamper each step causing a loss of efficiency that might be critical.”
We have corrected it.
Lines 80-81 please rewrite. Please explain what is “gammacel.” Is this the source of radiation? Should this be Gammacell? Also please change line 81 to “…irradiation sources applied to male flowers”
We have corrected it.
Lines 81 -88 should be “Although ionized pollen can germinate on the female stigma and grow pollen tubes to reach the embryo sac, they are unable to fertilize the egg-cell and the polar nuclei. Therefore, irradiated pollen stimulate egg-cell division and induce parthenogenetic embryos [3]. Overall, the dose of ionizing radiation can range from 25 to 500 Gy depending on the species and can yield less parthenogenetic haploid embryos at higher or more diploid embryos at lower dosages. Therefore irradiation should be optimized for each species because pollen sensitivity is attributed to radiation-resistance.”
We have corrected it.
Lines 104-107 are grammatically-incorrect and should be rewritten.
We have corrected it.
Lines 109-113 are grammatically-incorrect and should be rewritten.
We have corrected it.
Lines 126-130 are difficult to understand and grammatically-incorrect and should be rewritten.
We have rewritten those sentences.
Lines 141 should be “Colchicine was applied in vitro in solid E20H8 medium for 48h…”
We have changed it.
Lines 146-147 should be “Chromosome doubling is highly influenced by the genotype and the most suitable method must be determined empirically.”
We have changed it.
Lines 152-154 are difficult to understand and grammatically-incorrect and should be rewritten.
We have rewritten clearly.
Lines 182-184 require clarification. Do you mean that the vectors need to be tested in protoplasts before they are used for plant transformation? Do you just test for expression of the gRNA and Cas9 protein, or do you test whether the targeted gene is actually edited in protoplasts?
We have added clarification.
Lines 196-197 should be: Moreover, the transformation process is usually optimized for and restricted to a few genotypes. This impedes the application of transformation and genome editing to a wide range of genotypes of interest.
We have changed it.
Lines 255-256 “free-transgene” should be “transgene-free”
Corrected.
Line 267: what does “pollen-radio” mean?
Indeed, it is wrong. The correct way to say is radio-resistance of pollen.